# Vertically Aligned and Interconnected Graphite and Graphene Oxide Networks Leading to Enhanced Thermal Conductivity of Polymer Composites

**DOI:** 10.3390/polym12051121

**Published:** 2020-05-14

**Authors:** Ziming Wang, Yiyang Cao, Decai Pan, Sen Hu

**Affiliations:** College of Aerospace Engineering, Chongqing University, Chongqing 400044, China; 20176262@cqu.edu.cn (Y.C.); peter_spread@163.com (D.P.); 20176238@cqu.edu.cn (S.H.)

**Keywords:** polymer–matrix composite, graphene, thermal properties, mechanical testing

## Abstract

Natural graphite flakes possess high theoretical thermal conductivity and can notably enhance the thermal conductive property of polymeric composites. Currently, because of weak interaction between graphite flakes, it is hard to construct a three-dimensional graphite network to achieve efficient heat transfer channels. In this study, vertically aligned and interconnected graphite skeletons were prepared with graphene oxide serving as bridge and support via freeze-casting method. Three freezing temperatures were utilized, and the resulting graphite and graphene oxide network was filled in a polymeric matrix. Benefiting from the ultralow freezing temperature of −196 °C, the network and its composite occupied a more uniform and denser structure, which lead to enhanced thermal conductivity (2.15 W m^−1^ K^−1^) with high enhancement efficiency and prominent mechanical properties. It can be significantly attributed to the well oriented graphite and graphene oxide bridges between graphite flakes. This simple and effective strategy may bring opportunities to develop high-performance thermal interface materials with great potential.

## 1. Introduction

With the rapid development of electronic devices, thermal dissipation has become a critical necessity for its reliability, lifetime, and high speed [1,2,3]. To achieve an efficient management of thermal dissipation, thermal interface materials (TIM) are widely used in high-power electronics [4,5,6], especially in next-generation electronic devices such as the smart phone, high-performance computer, and light-emitting diodes. Polymer-based composites with high thermal conductivity play an irreplaceable role as a typical candidate of TIM [7,8]. That is not only because of their remarkable performance in thermal dissipation of electronics, but also their light-weight, flexibility, and easy processing features [9]. Usually, the inorganic thermal conductive fillers are uniformly dispersed in polymer matrix to obtain the high thermal conductivity, including metals [10,11,12], α-alumina (α-Al_2_O_3_) [13,14,15], hexagonal boron nitride (h-BN) [16,17,18,19], nanoclays [20], carbon nanotubes (CNTs) [21,22,23,24], and their hybrid mixtures [25,26]. However, with separated fillers, it is hard to introduce high thermal conductivity to the composites due to the lack of efficient heat transfer pathways and high interfacial thermal resistance between the fillers and polymer matrix [27]. Therefore, presently, some special strategies are also continuously developed, with constructing three-dimensional (3D) structures as one of the most effective methods.

Natural graphite flakes, as platelet-like thermally conductive fillers, have attracted lots of attention because of their high aspect ratio and theoretical thermal conductivity (~129 W m^−1^ K^−1^) [28]. It has been demonstrated that graphite can notably enhance the thermal conductive property of composites when they are added into the polymeric matrices [5,29,30]. Unlike special carbon materials such as graphene [31,32,33,34], carbon nanotubes [35,36], as well as fullerene [37], which have complex fabricating procedures, natural flake graphite is easily obtained. In addition, it is relatively simple to acquire high loading for graphite in a polymeric matrix without agglomeration via the usual fabrication ways due to its insensitivity to the van der Waals forces. Nevertheless, due to the intrinsic microstructure of graphite flakes, the thermal conductivity is anisotropic, that is, there is much higher thermal conductivity along the in-plane direction than along the through-plane direction [3]. Thus, the graphite flakes should be oriented in the heat transfer direction to fully utilize their in-plane thermal conductivity, instead of distributing them randomly.

To build the oriented architecture of anisotropic fillers, such as platelets [38,39], wires and fibers [40,41], inside the polymer matrix, many strategies have been adopted in recent years. For example, the vacuum filtration method was skillfully developed and utilized to achieve the oriented structures of thermal conductive fillers [26,42]. This method always shows notable availability for fabricating the paper-like composites [43] and obtaining high value of thermal conductivity in the horizontal direction. However, in many cases, prominent thermal transfer along the vertical direction is more desirable for realizing the efficient heat removal of TIM. Magnetic alignment is an attractive approach in this regard, in which fillers are aligned along the external magnetic field after coating magnetic iron oxide nanoparticles around the surface of the fillers [44,45]. However, the incorporated iron oxide limits the thermal conductivity of the composite fillers and increases the total mass. Meanwhile, other approaches, including hot-pressing [18,46], injection molding [47,48,49], 3D printing [50,51], and electrospinning [52], are also widely applied. The ice-templating self-assembly method, which can construct well-aligned architecture along the ice-growth direction, has been regarded as one of the most promising strategies [3,53,54]. Such method, however, requires the strong interaction between particles or platelets. Due to the lack of functional groups on the surface of graphite flakes, the interactions between them are so weak that the bridge and support should be well constructed.

Graphene oxide (GO), as one of the oxygen-containing derivatives of graphene, has attracted much attention because of its outstanding physical properties [55,56,57,58]. Meanwhile, there are superb hydrogen bonding interactions resulting from the abundant oxygen-containing groups (e.g., hydroxyl, carboxyl, and epoxide groups) on its basal planes and edges [59,60,61]. Recently, it has been clearly demonstrated that GO can interconnect with each other and assemble into a stable and specific structure by the ice-templating self-assembly method [1,54,62,63,64]. Thus, it presents significant potential for serving as the bridge and support of graphite flakes and forming 3D graphite networks. 

Herein, the vertically aligned graphite and GO networks were successfully synthesized by freeze-casting method with tuned temperatures. Graphite flakes were well oriented in the direction with their high thermal conductivity, and GO connected graphite flakes, presenting effective bridge and support actions. Furthermore, the thermal conductivity and mechanical properties were reasonably evaluated and analyzed, especially the impacts of the extremely low freezing temperature.

## 2. Materials and Experiments

### 2.1. Materials

Natural graphite flakes were provided by Sinopharm Chemical Reagent Co., Ltd., Shanghai, China. Hydroxyl-terminated polybutadiene (HTPB, 99.8%) was supplied by Hongyuan Chemical Industry & New Material Technology Co., Ltd., Shenzhen, China, and regarded as the prepolymer of polyurethane (PU). Isophorone diisocyanate (IPDI, 99%) and triphenylbismuthine (TPB) were obtained from Aladdin Co., Shanghai, China. Dibutyl phthalate (DBP, 99.5%) was provided by Xilong Scientific Co., Ltd., Shantou, China.

### 2.2. Preparation of 3D Graphite and Graphene Oxide Networks

Graphene oxide (GO) was fabricated from natural graphite flakes using modified Hummer’s method [65], and the concentration of its water suspension was tuned to 6 mg/mL. Then, this water suspension (10 mL) was mixed with the natural graphite flakes (3 g) by rapid stir for 30 min. Subsequently, the obtained homogeneous mixture was poured into a mold, followed by adequate freezing with three different substrate temperatures (−5 °C, −60 °C, and −196 °C). After freeze-drying at low temperature (−50 °C) for 36 h, aligned 3D graphite and graphene oxide (3D-GP) aerogels were finally obtained.

### 2.3. Preparation of Oriented 3D-GP/PU Network Composites

HTPB prepolymer with 25 wt % DBP, 7.5 wt % IPDI, and an appropriate amount of TPB was uniformly mixed through a mechanical agitator, and degassed in vacuum for 30 min at room temperature. The mixture was then infused into 3D-GP skeletons, and after that, the compounds were put into the vacuum oven at −25 Pa for 1 h. Because of the capillary action and low-pressure environment, the liquid HTPB prepolymer was successfully infiltrated into 3D-GP skeletons. Finally, the 3D-GP/PU composites were achieved after a curing process at 80 °C for 48 h in an electrothermal blowing dryer. The schematic illustration of the fabrication process is shown in Figure 1. For comparison, graphite flakes were directly mixed with HTPB prepolymer without 3D architecture, and the obtained composite was denoted as random GP/PU.

### 2.4. Characterization

Fourier transform infrared (FTIR) spectra of the samples were recorded on an infrared spectrophotometer (FT/IR-4100, Jasco, Tokyo, Japan). X-ray diffraction (XRD) patterns were obtained on an X-ray diffractometer (D/max-2500/PC, Rigaku, Tokyo, Japan) with Cu Kα radiation (λ = 1.5418 Å) at a scanning speed of 10°/min from 3° to 90°. Microstructure and morphology of the materials were performed with field-emission scanning electron microscope (SEM, SU-8010, Hitachi, Tokyo, Japan) and transmission electron microscope (TEM, JEM-2100F, JEOL, Tokyo, Japan) with an acceleration voltage of 200 kV. The uniaxial tensile measurement was performed on Instron 5980 with the loading rate of 10 mm/min. The thermal decomposition was performed with thermogravimetric analysis (TGA Q600 SDT, TA Instrument, New Castle, DE, USA) at the nitrogen atmosphere from room temperature to 800 °C at a heating rate of 10 °C/min. Thermal conductivity (*K*) was calculated using
*K=αρC_p_*(1)
where *α* is the thermal diffusivity coefficient obtained by LFA 467 HyperFlash instrument (Netzsch, Bavaria, Germany), *ρ* is the density of the composites measured by XS105DU automatic density analyzer (METTLER TOLEDO, Zurich, Switzerland), and *C_p_* is the specific heat capacity measured by DSC Q20 differential scanning calorimetry instrument (TA Instrument, New Castle, DE, USA).

## 3. Results and Discussion

### 3.1. Morphologic and Structural Characterization

Figure 2a shows the FTIR spectra results of graphite and GO. It can be seen that several types of functional groups were attached to the surface of GO without existing on graphite flakes. The presence of absorption peaks at 1726 cm^−1^ (C=O stretching vibration), 1627 cm^−1^ (C=C stretching vibration), 1168 cm^−1^ (C−O−C stretching vibration), 1037 cm^−1^ (C−OH stretching vibration), and 3300–3000 cm^−1^ (O−H stretching vibration) in the spectrum of GO indicates that graphite was successfully oxidized into GO by Hummer’s method. Because of the abundant functional groups on the surfaces and the hydrogen-bonding interactions between them simultaneously, it was feasible to construct 3D architecture via utilizing GO as the support of graphite flakes.

XRD patterns of graphite and GO are presented in Figure 2b. Graphite exhibited a sharp peak centered at 2θ = 26.4°, while in the pattern of GO, this peak could not be found and a characteristic peak at 2θ = 11.0° appeared. Thus, GO shows a notably larger interlayer distance than that of graphite, which clearly reveals the carbon layer exfoliation of graphite.

In addition, the SEM image of graphite flakes is shown in Figure 2c, and we can see that they exhibited a typical platelet-like shape with relatively high thickness. Meanwhile, the morphology of GO sheets was characterized by TEM observation, as shown in Figure 2d. It is clear that GO sheets presented much smaller thickness according to the excellent transmittance. The fold profile can also be found in the TEM image, which is one of the characteristic morphology features of GO sheets. This thin and flexible profile that is quite different from graphite makes GO sheets to be considered as the reasonable bridge of graphite flakes. More than this, GO possesses the prominent mechanical properties that bring the 3D structure more robustness and stability.

Figure 3a–c present the top-view digital images of 3D-GP skeletons with freezing temperatures of −5 °C, −60 °C, and −196 °C, respectively, and for simplicity, the samples were named with the freezing temperature. For example, 3D-GP5 means the skeleton with the freezing temperatures of −5 °C, and thus, the 3D-GP samples can be spelled into 3D-GP5, 3D-GP60, and 3D-GP196, respectively. We can see that the top-view images show different surface features. With the decreasing of freezing temperatures, the top surfaces of the skeletons were increasingly fine and smooth. That is because the ice crystals would be smaller and smaller with the reducing of freezing temperature. According to the previous study, the platelets of graphite and GO should be gradually driven to the gaps of ice crystals during the freezing procedure [53]. Consequently, we can dominate the skeleton compactness effectively by controlling the growth of ice crystals using temperature. Figure 3d–i show the side-view SEM images of 3D-GP skeletons and can further verify this elaboration. It also can be seen that 3D-GP skeletons were directionally organized, and graphite flakes were vertically assigned with GO serving as bridge and support. Figure 3d shows the profile of 3D-GP5, which exhibited typical hierarchically ordered and interconnected network of graphite and GO. The graphite flakes distributed along GO support and established the quasi parallel composite layers (Figure 3e). The layers were hierarchically arranged to construct a 3D structure. This phenomenon can also be found in 3D-GP60 and 3D-GP196 samples, which are shown in Figure 3f,h. However, due to the effect of ice crystal growing, the distances of vertical layers reduced with decreasing temperature. That is to say, lower temperature could achieve denser structure. Moreover, most graphite flakes dispersed vertically, and the magnifications reveal that they were arrayed orderly, resulting from the fabrication process.

After infiltration of HTPB prepolymer, this 3D graphite and GO network was embedded into the HTPB-based PU matrix, and a typical composite with vertically oriented internal structure was efficiently constructed. The morphology of the 3D-GP/PU composites is shown in Figure 4b–d. It is evident that graphite flakes and GO sheets dispersed directionally in PU matrices. Nevertheless, the distance of the oriented graphite flakes suggests clear relevance to temperature, because of the different sizes of ice crystal, which has been discussed previously. Resulting from the denser network of graphite and GO, the 3D-GP196/PU composite possessed the most uniform and tightest filler distribution (Figure 4d). In contrast, the random GP/PU composite without aligned graphite and GO network exhibited a rough and disordered surface (Figure 4a). This is dramatically different from 3D-GP/PU composites, where the irregular graphite flakes presented random distribution directions and discontinuous fillers in the PU matrix.

### 3.2. Thermal Performance of the Composites

In order to achieve the effects of 3D-GP skeletons on heat transfer, thermal conductivity values of random GP/PU, 3D-GP5/PU, 3D-GP60/PU, and 3D-GP196/PU composites were carefully measured. It is obvious that the anisotropic internal microstructure of the composites could lead to anisotropic thermal conductivity values at the different directions. Figure 5a shows the illustration for in-plane and out-of-plane directions, which are vertical and parallel to the graphite and GO layers that are described previously. Thermal conductivity of these two directions are presented in Figure 5b,c, respectively. We can clearly see that random GP/PU composite had a lower thermal conductivity value of 0.97 W m^−1^ K^−1^ at the weight loading of 33%, and it possessed the same thermal conductivity in in-plane and out-of-plane directions. Meanwhile, the thermal conductivity values of 3D-GP5/PU, 3D-GP60/PU, and 3D-GP196/PU composites were 1.01 W m^−1^ K^−1^, 1.18 W m^−1^ K^−1^, and 1.48 W m^−1^ K^−1^ in in-plane direction, and 1.46 W m^−1^ K^−1^, 1.74 W m^−1^ K^−1^, and 2.15 W m^−1^ K^−1^ in out-of-plane direction, respectively, at the same filler loading. On one hand, 3D-GP/PU composites occupied higher thermal conductivity than that of random GP/PU in both in-plane and out-of-plane directions. More precisely, 3D-GP196/PU had the highest thermal conductivity, and it increased by 52.6% and 121.6%, in in-plane and out-of-plane directions, respectively, compared with random GP/PU. On the other hand, the thermal conductivity values in out-of-plane direction of 3D-GP/PU composites were always higher than that in in-plane direction. For example, the out-of-plane thermal conductivity of 3D-GP196/PU was 45.3% higher than in-plane thermal conductivity, as calculated using the measured results. 

Moreover, according to the tested results, thermal conductivity of pure PU, which is the matrix of all the composite samples, was 0.2 W m^−1^ K^−1^. Thermal conductivity enhancement (TCE) can be further calculated and is shown in Figure 5d. Here, TCE is defined as
(2)$$\mathrm{TCE}=\frac{K_{c}-K_{m}}{K_{m}}\times 100\%$$
where *K_c_* and *K_m_* are the thermal conductivity of composites and pure PU matrix, respectively. It can be seen that the TCE of 3D-GP196/PU composite was 640% and 975% in in-plane and out-of-plane, respectively, which are much higher than that of random GP/PU, 3D-GP5/PU, and 3D-GP60/PU composites. Consequently, 3D-GP196 obviously had the most effective action on improving the thermal conductivity of PU matrix.

The thermal conductivities (TC) of the composites with temperature are shown in Figure 6a,b. From the results, both the out-of-plane and in-plane TC values present minor decrease with the increasing of measurement temperature. That indicates the stable thermal dissipation of the composites, which is of benefit to the long-term thermal management of devices. 

In order to compare with other kinds of thermal conductive composites, TC enhancement efficiency (*η*) is regarded as an important index, which can be defined as TC enhancement per 1 wt % loading, which is given as follows
(3)$$\eta =\frac{K_{c}-K_{m}}{100WK_{m}}\times 100$$
where *η* is the TC enhancement efficiency and *W* is the weight loading of fillers. The *η* values of the reported composites are summarized in Figure 6c and Table 1. Obviously, the aligned graphite/GO/PU composite exhibited the highest *η*, which means the highest TC at the same loading, across all the composites.

Two principal reasons should be mentioned that resulted in the significant enhancement of thermal conductivity of the composites. The first is that most of the graphite flakes lay along the direction that occupied higher thermal conductivity (Figure 7a). Thus, the anisotropic thermal performance of graphite was efficiently utilized, and in this direction, we could achieve relatively high thermal conductivity. The second is that GO sheets served as the bridges of graphite flakes, which were also the excellent bridges of heat transfer pathways (Figure 7b). It means that the 3D-GP/PU composites had continuous heat transfer channels composed of graphite and GO simultaneously. Nonetheless, the random GP/PU had no helpful impact on the GO bridge. 

In addition, it also can be noticed that the more thermal transfer pathways the composite had, the higher thermal conductivity it occupied, which was considerably useful for heat transport. In these four composites, 3D-GP196/PU possessed the densest filler structure benefiting from the architecture of 3D-GP196 skeleton. It essentially depended on the extremely low freezing temperature and the smallest ice crystals.

TGA measurements were performed to investigate the thermal stability of composites as shown in Figure 8. It can be seen that 3D-GP196/PU presented the highest thermal stability, related to the close bonding between fillers and PU, which impeded the movement and decomposition of PU molecular chains. Compared with 3D-GP196/PU, the other composites showed worse performance, and the composite with lower freezing temperature exhibited better thermal stability. It can be attributed to the smaller ice crystals, which is considered to result in the relatively low freedom state of molecular chains.

### 3.3. Mechanical Properties of 3D-GP/PU

Figure 9 shows the tensile properties of pure PU and its composites fabricated in this work. Because of special internal structure of the composites, the tensile direction is vertical to the graphite and GO layer. It is clear that the composites, including random GP/PU and 3D-GP/PU, exhibited dramatically varied mechanical performances (Figure 9a). For example, random GP/PU had a relatively high elastic modulus, compared with 3D-GP60/PU and 3D-GP196/PU (Figure 9b). That is caused by the weak connect of 3D-GP60 and 3D-GP196 skeletons in the tensile direction. Although 3D-GP5 skeleton also possesses this weak connect, the elastic modulus of 3D-GP5/PU was higher than random GP/PU. It might have resulted from the stress concentration in this composite, owing to the gathering and non-uniform phenomenon in the 3D-GP5 skeleton. Because of that, its tensile strength and elongation at break were much lower than random GP/PU and 3D-GP196/PU (Figure 9c,d). As we know, 3D-GP196 skeleton is more homogenous, and 3D-GP196/PU composite can occupy relatively high strength of 0.39 MPa and elongation of 341%. Its strength was enhanced by 40.6% compared to pure PU, and was higher than that of 3D-GP5/PU and 3D-GP60/PU. Meanwhile, its high elongation at break clearly suggests the nice flexibility of this composite. Thus, the uniform graphite and GO network fabricated via extremely low temperature results in prominent mechanical properties. It is very significant for the application of TIM.

## 4. Conclusions

In summary, the graphite and GO networks were prepared through a freeze-casting method using three freezing temperatures, and most graphite flakes lay along vertical direction with GO serving as the bridge and support. The 3D-GP skeleton with low freezing temperature of −196 °C and its PU-based composite occupied a more uniform and denser structure. As a result, the thermal conductivity of 3D-GP196/PU reached 1.48 W m^−1^ K^−1^ and 2.15 W m^−1^ K^−1^, and increased by 52.6% and 121.6%, in in-plane and out-of-plane directions, respectively, compared with random GP/PU. Moreover, this composite presented high thermal conductivity enhancement efficiency and thermal stability. It was demonstrated that the phenomenon can be attributed to the well-aligned graphite and the effect of GO bridges between graphite flakes. Meanwhile, the uniform 3D-GP196 skeleton can also result in excellent mechanical properties. These thermal and mechanical performances are requisite for ultimate thermal management applicability. This simple and effective approach provides a promising strategy to develop high performance TIM.

## Figures and Tables

**Figure 1 polymers-12-01121-f001:**
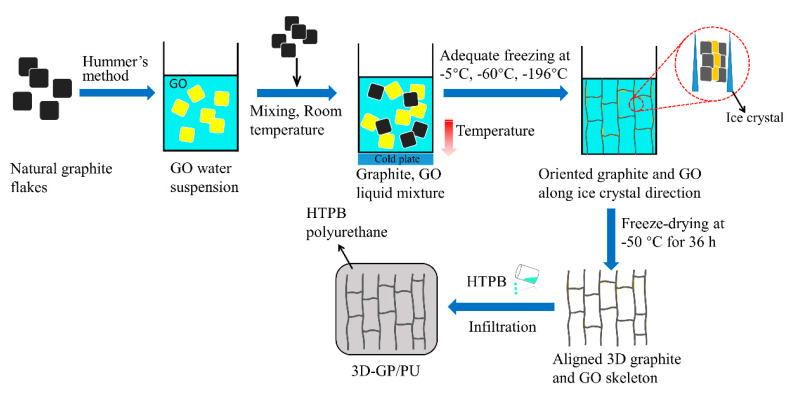
Schematic diagram of the fabrication process of 3D-GP/PU composites.

**Figure 2 polymers-12-01121-f002:**
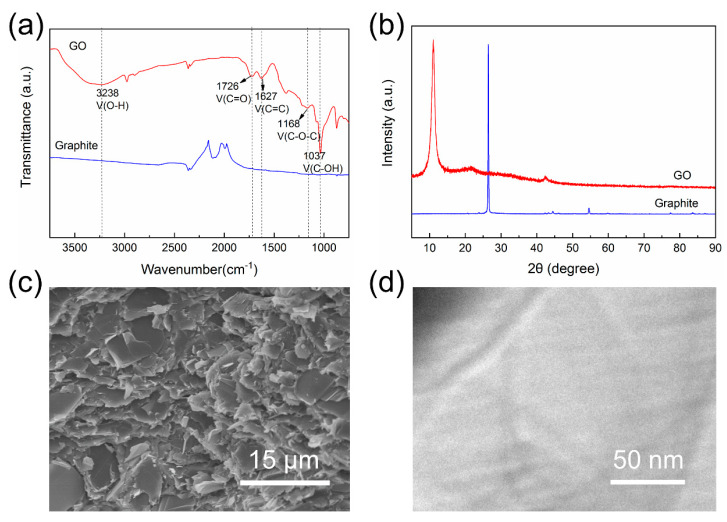
(**a**) FTIR spectra of graphite and graphene oxide (GO). (**b**) XRD patterns of graphite and GO. (**c**) SEM image of graphite flakes. (**d**) TEM image of GO.

**Figure 3 polymers-12-01121-f003:**
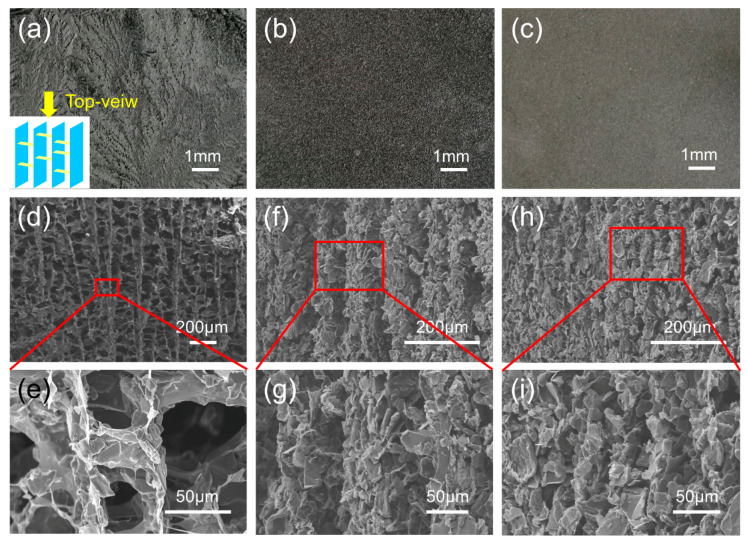
(**a**–**c**) Top-view digital images of 3D-GP skeletons with freezing temperatures of −5 °C, −60 °C, and −196 °C, respectively. Inset is the illustration of top-view direction. (**d**–**i**) Side-view SEM images of 3D-GP skeletons and their magnifications with freezing temperatures of −5 °C, −60 °C, and −196 °C, respectively.

**Figure 4 polymers-12-01121-f004:**
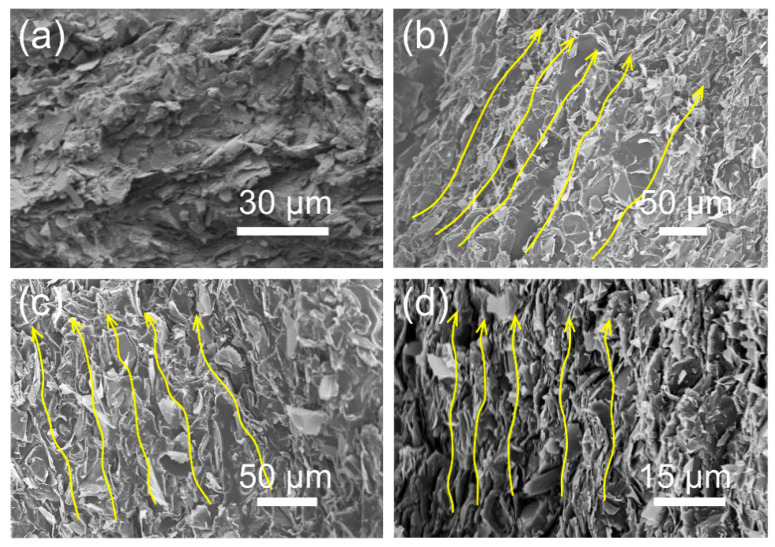
SEM images of (**a**–**d**) random GP/PU, 3D-GP5/PU, 3D-GP60/PU, and 3D-GP196/PU composites, respectively. Yellow arrows indicate the direction of graphite and GO.

**Figure 5 polymers-12-01121-f005:**
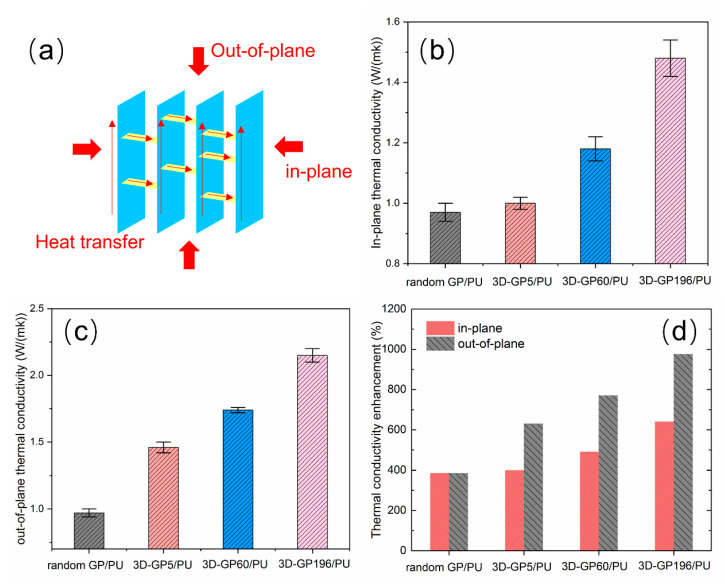
(**a**) Illustration for in-plane and out-of-plane directions and heat transfer pathways. (**b**,**c**) In-plane and out-of-plane thermal conductivities of the composites, respectively. (**d**) Thermal conductivity enhancement of the composites compared with the pure PU.

**Figure 6 polymers-12-01121-f006:**
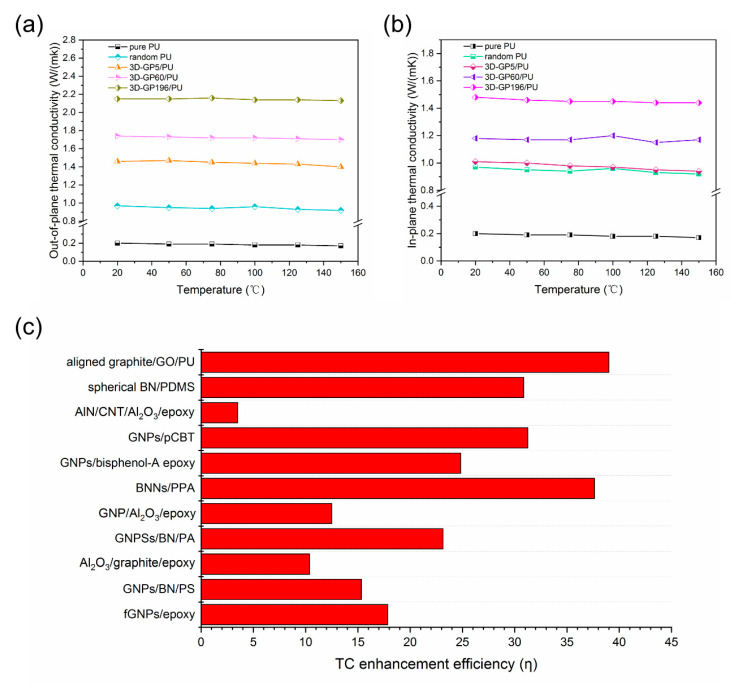
(**a**) Out-of-plane and (**b**) in-plane thermal conductivity with temperature. (**c**) Comparison of thermal conductivity (TC) enhancement efficiency of reported composites.

**Figure 7 polymers-12-01121-f007:**
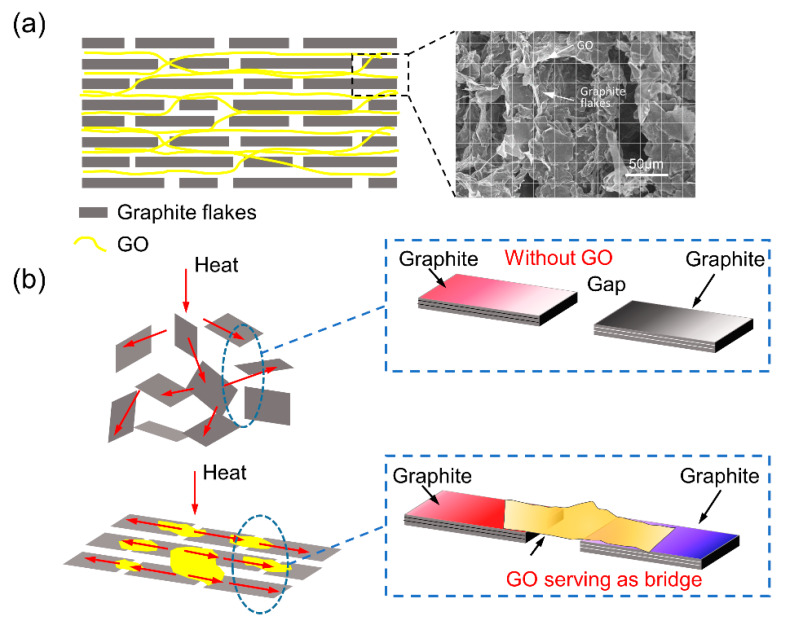
(**a**) Cross-section of the graphite and GO layer, indicating the arrangement. The right graph is the SEM cross-section image. (**b**) Schematic of the thermal conductive channel of random GP/PU and 3D-GP/PU with GO serving as bridge.

**Figure 8 polymers-12-01121-f008:**
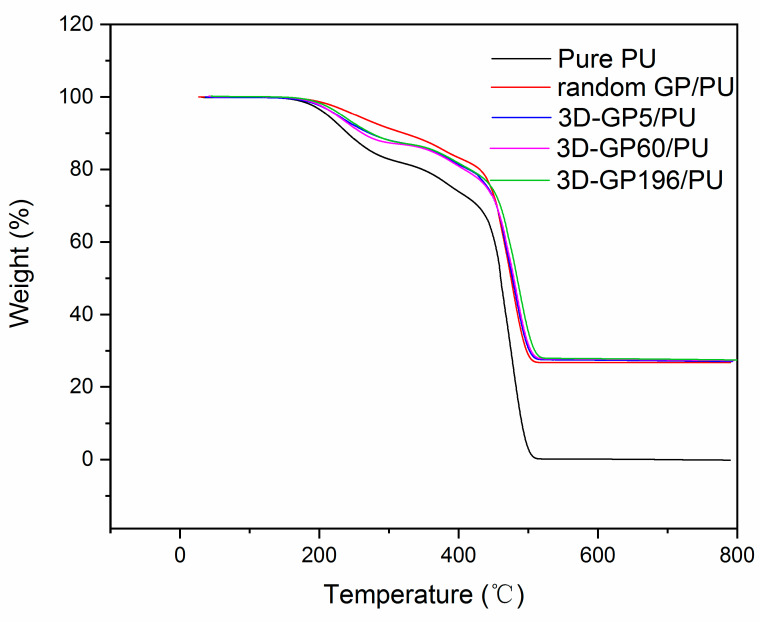
TGA curves of PU and composites.

**Figure 9 polymers-12-01121-f009:**
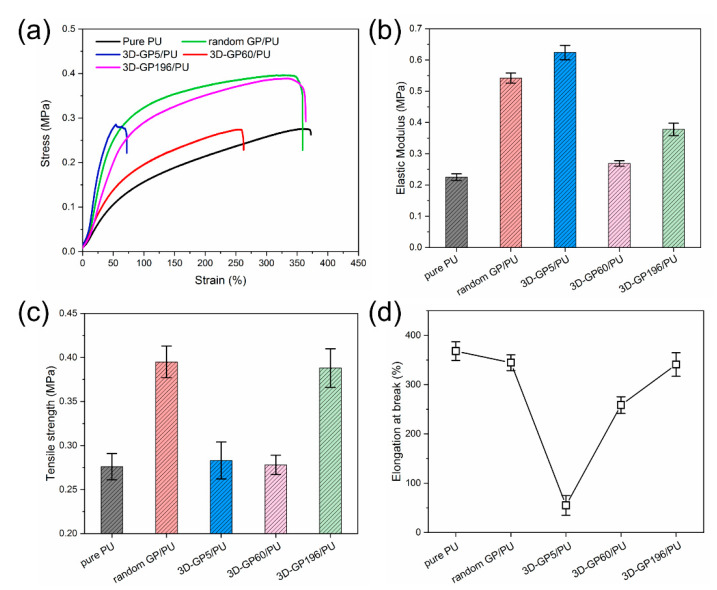
(**a**) Tensile stress-strain curves of the composites. (**b**–**d**) Elastic modulus, tensile strength, and elongation at break of the composites, respectively.

**Table 1 polymers-12-01121-t001:** Summary of the TC of polymer composites filled with different fillers.

Sample	TC of Matrix (W m^−1^ K^−1^)	TC of Composites (W m^−1^ K^−1^)	Loading (wt %)	TC Enhancement Efficiency (*η*)	References and Years
fGNPs/epoxy	0.23	1.49	30.00	17.85	2017 [66]
GNPs/BN/PS	0.16	0.67	21.50	15.33	2015 [67]
Al_2_O_3_/graphite/epoxy	0.22	0.64	18.40	10.38	2015 [68]
GNPSs/BN/PA	0.28	1.69	21.5	23.14	2015 [67]
GNP/Al_2_O_3_/epoxy	0.20	2.2	80.00	12.50	2016 [69]
BNNs/PPA	0.18	2.89	40.00	37.63	2019 [19]
GNPs/bisphenol-A epoxy	0.20	1.70	30.00	24.83	2016 [70]
GNPs/pCBT	0.24	2.49	30.00	31.25	2016 [34]
AlN/CNT/Al_2_O_3_/epoxy	0.2	0.55	50.00	3.50	2015 [71]
spherical BN/PDMS	0.14	2.3	50.00	30.85	2019 [72]
aligned graphite/GO/PU	0.2	2.15	25.00	39.00	this work

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
