# Peer review of "Vertically Aligned and Interconnected Graphite and Graphene Oxide Networks Leading to Enhanced Thermal Conductivity of Polymer Composites"

_polymers, 2020, doi:10.3390/polym12051121_

Round 1

Reviewer 1 Report

The present manuscript entitled "Vertically Aligned and Interconnected Graphite and Graphene Oxide Networks Leading to Enhanced Thermal Conductivity of Polymer Composites'' describes the construction of vertically aligned and interconnected uniform graphite frames using graphene oxide as linker and freeze-casting method to shape the final structure. The formed network was filled in polymeric matrix and the final composite exhibited excellent thermal properties and improved mechanical properties.

This paper is well-demonstrated and the conclusions of the study are completely supported by the results. I consider it would be an excellent addition in Polymers and thus I recommend its publication with only a minor correction. In Figure 7a, the arrows are not well located to show exactly GO and graphite. In this context, I suggest to add some transparent grids in the SEM image, in order to better display the location of GO and graphite. Finally, add in the caption that it is a SEM cross-section image.

Reviewer 2 Report

I have carefully read the manuscript entitled Vertically Aligned and Interconnected Graphite and Graphene Oxide Networks Leading to Enhanced Thermal Conductivity of Polymer Composites. The manuscript presents an interesting topic – composites based on aligned and interconnected graphite and graphene oxide for enhanced thermal conductivity. The manuscript presents an innovative strategy in order to obtain high thermal conductivity-based composites, the filler being a network of oriented graphite and graphene oxide. The manuscript is very well written, the synthesis strategy is very well organized and the results are very well discussed (all the graphs present error bars, the spectra are indexed and the SEM are indicated precisely on the micrographs with interesting zones). Also, a comparison with previous obtained results in literature is presented in Table 1. The Conclusions are suitable for presented work. The references are very well organized covering a wide range of reports previously reported.

Reviewer 3 Report

The manuscript deals with the assessment of improvement in thermal conductivity of characteristics of polymer composites reinforced with vertically aligned graphitic materials. It is a work of definite technical interests with excellent writing and presentation. However, it needs a moderate revision and details to be added, to be accepted. Specific comments are given below:

  1. Given that, it is a tedious process to align graphitic materials vertically to improve thermal conductivity, how viable is the synthesis for the overall performance of the composites for any application?
    1. How about the mechanical properties, electrical properties?
    2. What was the ultimate applicability of such composites?
    3. Include some approximate cost analysis?

These details are missing and I believe including them would really heighten the value of this work.

  1. FTIR and XRD:
    1. Provide peak assignments to graphite as well.
    2. Where are the FTIR spectra for the composites?
    3. Where are the XRD patterns of the composites?

  1. Figure 2(d) – What do we really learn from this figure. It is almost non-illustrative!

  1. Has the author tried or assessed the effects of the diagonal arrangement of graphitic materials?

  1. While I say that the manuscript is technically well written, it generally has a lot of grammatical errors, inappropriate punctuations, and inappropriate use of capital letters. Particularly, tenses change quite abruptly. Please take care of these in the revision.

Round 2

Reviewer 3 Report

The manuscript is revised to the satisfaction. The language should be checked properly before publication which is the only persisting issue.